# RANGE-NULL LATENT PRIOR-GUIDED CONSISTENCY MODEL FOR LOW LIGHT IMAGE ENHANCEMENT

## ABSTRACT

Low light image enhancement (LLIE) is a challenging task, with most existing models often struggling to adapt to diverse dark environments due to insufficient training datasets. In this paper, we propose a novel unsupervised model called **R**ange-null **L**atent **P**rior-guided **C**onsistency **M**odel (**RLPCM**), which integrates a latent consistency model (LCM) into low light enhancement using Retinex-based range-null space decomposition. RLPCM leverages an off-the-shelf LCM as a generative prior to improve both the latent consistency and realness of enhanced images. Meanwhile, fine-tuning a lighting decoder solely on normal-light images to ensure high fidelity in image space. A key contribution is a simple yet effective global illumination adjustment applied to the range-space component, along with a natural language guidance module to learn the null-space component. This allows for iterative generation to enhance both consistency and realness in just a few steps. Additionally, we present a new UAV low light dataset (UAV-LL) containing 300 image pairs from various UAV scenarios to support comprehensive evaluation. Extensive experiments demonstrate the superior adaptability and effectiveness of our framework across a wide range of low-light environments.

## 1 INTRODUCTION

Low light image enhancement (LLIE) is a long-standing problem that influences both human visual perception and related computer vision tasks, such as depth estimation Wang et al. (2021), object detection Hashmi et al. (2023), and semantic segmentation Pan et al. (2024). Current LLIE approaches are generally categorized into two types: supervised and unsupervised methods. Supervised methods Weng et al. (2024); Cai et al. (2023); Jiang et al. (2023) typically rely on paired images to train end-to-end models, but acquiring pixel-aligned image datasets is particularly challenging, especially in mobile environments. On the other hand, unsupervised methods Ma et al. (2022); Yang et al. (2023); Liang et al. (2023) have gained traction by minimizing the need for paired data. Among these, diffusion-based models Wang et al. (2024); Jiang et al. (2024a) have attracted significant attention for their powerful generative capabilities. Therefore, in this paper, we focus on diffusion-based approaches for low light image enhancement.

Most diffusion-based models for the LLIE task have integrated low-light image and illumination as conditional inputs to preserve image details and enhance illumination quality, as shown in Figure 1(a). For instance, QuadPrior Wang et al. (2024) introduces an illumination-invariant prior as a conditional generative model utilizing ControlNet Zhang et al. (2023). Similarly, LightenDiffusion Jiang et al. (2024a) employs a Retinex-based diffusion model that works with unpaired images, decomposing them into reflectance and illumination maps. However, these approaches lack an effective latent prior to guide the conditional diffusion process, often resulting in suboptimal enhancement outcomes and time-consuming procedures.

Recently, the denoising diffusion null-space model (DDNM) Wang et al. (2023b) incorporated range-null space decomposition Schwab et al. (2019) into diffusion models to address various image restoration (IR) tasks, such as image super-resolution, colorization, and deblurring. This method identifies appropriate null-space components to ensure realistic results, while applying a specific degradation operator to preserve range-space content for data consistency, thus achieving effective performance. However, DDNM depends on explicit degradation priors, which are challenging to obtain for LLIE task, and is often limited by slow inference speed.

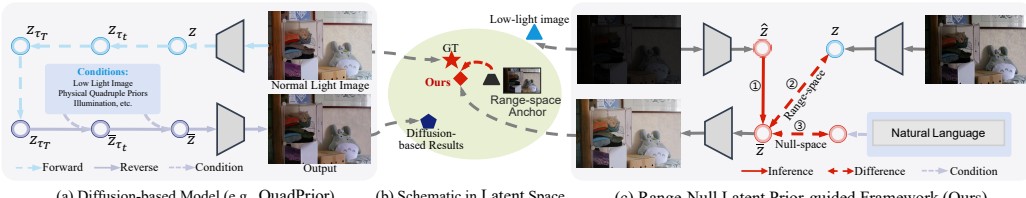

(a) Diffusion-based Model (e.g., QuadPrior)  (b) Schematic in Latent Space  (c) Range-Null Latent Prior-guided Framework (Ours)

Figure 1: (a) Existing diffusion-based models, such as DiffLL, QuadPrior, and LightenDiff, require the diffusion network to be trained with specific conditions. (b) The schematic process shows that the range-space anchor serves as a more effective initial latent prior. By applying our updating rule, the result in latent space becomes closer to the ground truth (GT). (c) We introduce a range-null latent prior-guided framework for the LLIE task, featuring a training-free diffusion process.

To address these issues, we propose a novel unsupervised framework called **R**ange-null **L**atent **P**rior-guided (**RLP**) for the LLIE task, as illustrated in Figure 1(c). To accelerate the diffusion process, the latent consistency model (LCM) Song et al. (2023) is integrated into low-light enhancement by leveraging Retinex-based range-null space decomposition to identify relevant null-space content, referred to as the RLP Consistency Model (RLPCM) in Figure 2. Our approach integrates a physics-driven model, *i.e.*, Retinex theory Land & McCann (1971), with null-space decomposition to introduce global illumination degradation that guides the range-space content. RLPCM relies solely on a current off-the-shelf latent consistency model as the generative prior, fine-tuning a lighting decoder with normal light images to enhance fidelity in the image space. We introduce a simple yet effective global illumination prior that fixes the range-space component within the range-null space decomposition. Additionally, we design a natural language guidance mechanism to facilitate learning in the null-space, enabling a few-step iterative generation process that effectively and fast balances the latent consistency and the realness.

In this paper, our key findings and contributions are summarized as follows: **1)** To the best of our knowledge, we are the first to introduce a consistency model into the LLIE task, effectively bridging Rang-null space decomposition with the Retinex theory into the consistency model. Compared to existing diffusion methods, our approach exhibits superior flexibility and robust across diverse scenarios. **2)** We propose a range-null latent prior-guided framework for the LLIE task, featuring a simple yet efficient global illumination prior that physically guarantees the reliability of range-space content. this prior can be manually adjusted during inference, thus avoiding the irreversible effects associated with fixed parameters. Additionally, we incorporate language-aware guidance mechanisms to facilitate the learning of null-space content. **3)** We present a novel UAV low-light dataset (UAV-LL), comprising 300 image pairs captured in various mobile environments. This dataset allows for a comprehensive evaluation of the generalization capabilities of existing methods under previously unseen conditions. *For the related work, please refer to Appendix A.*

## 2 METHODOLOGY

### 2.1 PRELIMINARIES

**Retinex Theory.** Among physics-driven models fundamental to LLIE, Retinex theory Land & McCann (1971) stands out as a key approach. The vanilla Retinex theory assumes that a low-light image $\mathbf{y}$ can be decomposed into illumination $\mathbf{A}$ and reflectance $\mathbf{x}$. Typically, $\mathbf{x}$, being an invariant physical property, is regarded as the ideal enhanced outcome. This relationship is expressed mathematically by the Retinex theory as:

$$\mathbf{y} = \mathbf{A} \odot \mathbf{x}, \tag{1}$$

where $\odot$ denotes the pixel-wise multiplication. But, accurately decomposing illumination $\mathbf{A}$ to estimate reflectance $\mathbf{x}$ remains a ill-posed problem. Traditional methods Guo et al. (2017), are typically Retinex-based illumination optimization problem. Most current methods Weng et al. (2024); Cai et al. (2023) aim to train end-to-end models that directly map $\mathbf{x}$ to $\mathbf{y}$, bypassing the need for explicit illumination estimation. However, these methods often rely on hand-crafted priors or task-

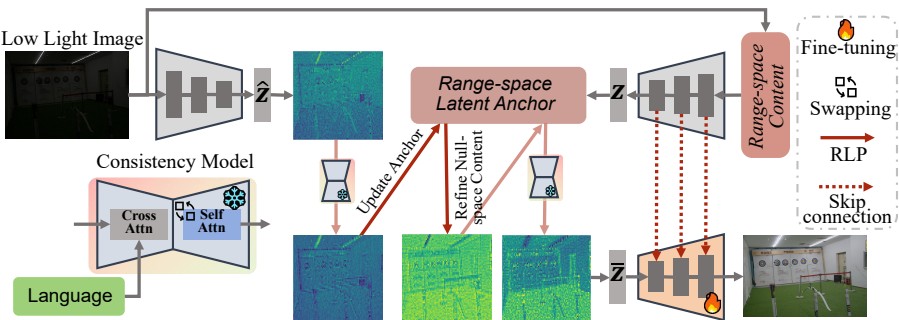

Figure 2: An overview of our RLP Consistency Model (RLPCM), based on the RLP framework in the Figure 1(c). The low-light image $\mathbf{y}$ and range-space content $\mathbf{A}^\dagger\mathbf{y}$ are both transformed into latent space via a pre-trained encoder, producing $\hat{\mathbf{z}}$ and $\mathbf{z}$, respectively. Here, $\mathbf{z}$ serves as a range-space anchor in the latent space, while the null-space content is refined using training-free conditional consistency sampling with language-aware attention swapping. Finally, a lighting decoder is fine-tuned with skip connections to preserve high fidelity, ensuring accurate reconstruction.

specific training, which limits their robustness and flexibility in diverse scenarios. Thus, integrating Retinex theory with pre-trained, off-the-shelf models presents a potentially promising solution.

**Range-null Space Decomposition.** offers a promising method to integrating Retinex theory with pre-trained, off-the-shelf models. Given a non-zero linear operator $\mathbf{A} \in \mathbb{R}^{mn \times mn}$, its pseudo-inverse $\mathbf{A}^\dagger \in \mathbb{R}^{mn \times mn}$ satisfies the equation $\mathbf{A}\mathbf{A}^\dagger\mathbf{A} = \mathbf{A}$. Hence, any sample $\mathbf{x} \in \mathbb{R}^{mn \times 1}$ can be decomposed into the range and null spaces of $\mathbf{A}$ as follows:

$$\mathbf{x} = \mathbf{A}^\dagger\mathbf{A}\mathbf{x} + \left(\mathbf{I} - \mathbf{A}^\dagger\mathbf{A}\right)\mathbf{x}, \tag{2}$$

where the first term $\mathbf{A}^\dagger\mathbf{A}\mathbf{x}$ represents the range-space content due to $\mathbf{A}\mathbf{A}^\dagger\mathbf{A}\mathbf{x} = \mathbf{A}\mathbf{x}$, and the second term represents the null-space content as $\mathbf{A}\left(\mathbf{I} - \mathbf{A}^\dagger\mathbf{A}\right)\mathbf{x} = 0$. Now, we reinterpret Retinex theory in the context of range-null space for a low-light image $\mathbf{y}$, aiming to derive the reflectance $\hat{\mathbf{x}}$ under the following constraints:

$$\text{Consistency: } \mathbf{A}\hat{\mathbf{x}} = \mathbf{y}, \quad \text{Realness: } \hat{\mathbf{x}} \sim p\left(\mathbf{x}\right), \tag{3}$$

where $p\left(\mathbf{x}\right)$ denotes the real reflectance distribution of $\mathbf{x}$. The general solution for reflectance $\hat{\mathbf{x}}$ that satisfies the consistency constraint is given by $\mathbf{A}\hat{\mathbf{x}} = \mathbf{y}$, leading to $\hat{\mathbf{x}} = \mathbf{A}^\dagger\mathbf{y} + \left(\mathbf{I} - \mathbf{A}^\dagger\mathbf{A}\right)\tilde{\mathbf{x}}$. The term $\tilde{\mathbf{x}}$ influences the realistic details in the null-space. Previous methods have sought to estimate the null-space content $\tilde{\mathbf{x}}$ using GANs Wang et al. (2023a) and diffusion models Wang et al. (2023b); Gandikota & Chandramouli (2024), but these methods inherit limitations such as the randomness inherent in diffusion models. In contrast, we observe that the consistency model Song et al. (2023) is more effective for generating null-space content due to its self-consistency property, while also providing faster inference speeds compared to existing diffusion models.

**Consistency Models.** Consistency models Song et al. (2023) are a novel class of generative models that have shown considerable potential across various vision tasks, including image generation and editing. Unlike diffusion models Ho et al. (2020); Song et al. (2021), consistency models enable single-step iterative generation, allowing for direct mapping from any point on the Probability Flow (PF) ODE trajectory back to its origin. Specifically, given a sequence of time points $\tau_i \in [\kappa, T]$, where $\tau_\kappa > \tau_{\kappa+1} > \cdots > \tau_T$, the solution trajectory $\{\mathbf{x}_{\tau_i}\}, \tau_i \in [\kappa, T]$ belongs to the PF ODE. The consistency function is defined as $f_\phi\left(\mathbf{x}_{\tau_i}, \tau_i\right) \to \mathbf{x}_{\tau_\kappa}$, and its self-consistency ensures that $f_\phi\left(\mathbf{x}_{\tau_{\tilde{\kappa}}}, \tau_{\tilde{\kappa}}\right) = f_\phi\left(\mathbf{x}_{\tau_{\hat{\kappa}}}, \tau_{\hat{\kappa}}\right)$ for $\tilde{\kappa}, \hat{\kappa} \in [\kappa, T]$. Recently, latent consistency models (LCM) Luo et al. (2023) have further enhanced efficiency by transforming $\mathbf{x}$ into latent space $\mathbf{z}$, resulting in improved computational performance. LCMs can be trained by distilling pre-trained diffusion models. In this work, we use the following LCM parameterization:

$$f_\phi\left(\mathbf{z}, c, \tau_i\right) = s_\kappa\left(\tau_i\right)\mathbf{x} + s_{\text{out}}\left(\tau_i\right)\hat{\mathbf{z}}_0 \tag{4}$$

$$\hat{\mathbf{z}}_0 = \left(\frac{\mathbf{z}_{\tau_i} - \sigma\left(\tau_i\right)\epsilon_\phi\left(\mathbf{z}, c, \tau_i\right)}{\alpha\left(\tau_i\right)}\right), \tag{5}$$

where $s_\kappa\left(\tau_i\right)$ and $s_{\text{out}}\left(\tau_i\right)$ are differentiable functions specifically defined as $s_\kappa\left(\tau_i\right) = 0$ and $s_{\text{out}}\left(\tau_i\right) = 1$, and $\epsilon_\phi\left(\mathbf{z}, c, \tau_i\right)$ is the teacher diffusion model, whose forward process can be effectively expressed as $\mathbf{z}_{\tau_i} = \alpha\left(\tau_i\right)\mathbf{z}_0 + \sigma\left(\tau_i\right)\epsilon_\phi$.

## 2.2 OUR APPROACH

Our goal is to establish a credible range-space that produces consistent results while accurately identifying a suitable null-space to enhance realness. Unlike the super-resolution and deblurring tasks that can define explicit degradation operators, LLIE is affected by variable illumination conditions, which complicates the design of a unified degradation operator. Building upon our RLP framework illustrated in the Figure 1(c), we introduce a Range-Null Latent Prior-guided Consistency Model (RLPCM), as depicted in Figure 2. RLPCM leverages Retinex theory to propose an adaptive and flexible illumination degradation factor derived from low-light images, thereby stabilizing the range-space component. Furthermore, we guide the null-space content through established consistency models by employing natural language for a more intuitive representation.

**Range-space Content Correction.** Given a low-light image $\mathbf{y}$, we first construct an illumination degradation factor $\mathbf{A}$, which will be discussed in detail in subsection 2.3, and then derive the range-space content using $\mathbf{A}^{\dagger}\mathbf{y}$. Subsequently, we employ an off-the-shelf latent consistency model $f_{\phi}(\cdot, \cdot, \cdot)$ Luo et al. (2023) and transform the low-light image $\mathbf{y}$ and range-space content $\mathbf{A}^{\dagger}\mathbf{y}$ into the latent space as $\hat{\mathbf{z}}$ and $\mathbf{z}$, respectively, via a pre-trained encoder. In this context, we consider $\mathbf{z}$ as the anchor representing range-space content in the latent space. Notably, the origin of the ODE trajectory of the LCM may be situated far from this anchor, and we thus maintain a fixed distance between the anchor and the output of the LCM as follows:

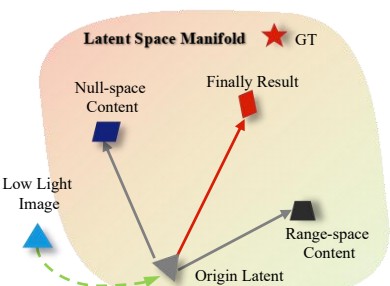

$$\epsilon = \mathbf{z} - f_{\phi}\left(\hat{\mathbf{z}}, c_{\text{null}}, \tau_i\right), \qquad (6)$$

where $\tau_i$ represents the time point and $c_{\text{null}}$ denotes the null condition. This distance $\epsilon$ is designed to bridge the gap, as illustrated in Figure 3. This bridging mechanism is essential for maintaining alignment between the generated output and the expected range-space content. The model's self-consistency

**Figure 3:** Schematic of range-null space decomposition in latent space. The dotted green line represents the trajectory of LCM. In the latent space, the range and null-space content are adjusted along the LCM's original trajectory, moving the position closer to the optimal result.

ensures that $\epsilon$ consistently points toward the anchor, thereby enhancing the overall coherence of the transformation process. Next, the key problem is to identify an appropriate null-space content that guarantees the realness results.

**Language-aware Null-space Content Refinement.** Inspired by Classifier-Free Guidance (CFG) Ho & Salimans (2022) for generating high-quality language-aligned images, we design a language-aware null-space content refinement module. Specifically, we utilize two contrasting language prompts to sample the conditional results, and the entire process can be articulated as follows:

$$\Delta\epsilon = f_{\phi}\left(\hat{\mathbf{z}}, c_n, \tau_i\right) - f_{\phi}\left(\hat{\mathbf{z}}, c_l, \tau_i\right), \qquad (7)$$

where $\Delta\epsilon$ denotes the null-space content aimed at improving the realness, $c_l$ denotes low-light language prompt, $c_n$ denotes normal-light language prompt. Recall the CFG Ho & Salimans (2022) employed in the LCM Luo et al. (2023), which involves replacing the original noise prediction with a linear combination of both conditional and unconditional noise derived from the teacher diffusion model, expressed as: $\tilde{\epsilon}_{\theta}(\mathbf{z}_{\tau_i}, \omega, c, \tau_i) := (1 + \omega)\epsilon_{\theta}(\mathbf{z}_{\tau_i}, c, \mathbf{z}_{\tau_i}) - \omega\epsilon_{\theta}(\mathbf{z}_{\tau_i}, c_{\text{null}}, \mathbf{z}_{\tau_i})$ and $\omega$ is called the guidance scale. We can rewrite equation 7 as follows:

$$\Delta\epsilon = (1 + w_1) f_{\phi}\left(\hat{\mathbf{z}}, c_n, \tau_i\right) - (1 + w_2) f_{\phi}\left(\hat{\mathbf{z}}, c_l, \tau_i\right) + (w_2 - w_1) f_{\phi}\left(\hat{\mathbf{z}}, c_{\text{null}}, \tau_i\right), \qquad (8)$$

By combining equation 6 and equation 8, therefore, we can derive the complete range-null space decomposition result as follows:

$$\bar{\mathbf{z}} = \epsilon + f_{\phi}\left(\hat{\mathbf{z}}, c_{\text{null}}, \tau_i\right) + \left(\mathbf{I} - \gamma\mathbf{A}^{\dagger}\mathbf{A}\right)\Delta\epsilon \quad \longrightarrow \quad \bar{\mathbf{z}} = \mathbf{z} + \left(\mathbf{I} - \gamma\mathbf{A}^{\dagger}\mathbf{A}\right)\Delta\epsilon \qquad (9)$$

where we set $\gamma \in [0, 1]$ to prevent $\left(\mathbf{I} - \gamma\mathbf{A}^{\dagger}\mathbf{A}\right) \equiv 0$, thereby enhancing the null-space content. Additionally, providing an accurate language condition poses a challenge in preserving texture while improving illumination for Low-Light Image Enhancement (LLIE). Inspired by MasaCtrl Cao et al. (2023) and Infedit Xu et al. (2024), we employ a swapping self-attention mechanism that facilitates non-rigid semantic transformations for image style transfer. This approach allows for querying local

---

**Algorithm 1** Range-Null Latent Prior-guided Consistency Model for LLIE

---

1: **Input:** Low light image $\mathbf{y}$, Global illumination intensity $\varpi$, Low light prompt $c_l$, Normal light prompt $c_n$, language guidance scale $w_1$ and $w_2$, time-step scheduler $\tau_T$, off-the-shelf LCM $f_\phi(\cdot, \cdot, \cdot)$ and encoder $\mathcal{E}(\cdot)$ and fine-tuned lighting decoder $\mathcal{D}(\cdot)$
2: **Output:** Enhanced image $\hat{\mathbf{x}}$
3: $\mathbf{A} = \varpi\mathbf{I}, \mathbf{A}^\dagger = \frac{1}{\varpi}\mathbf{I}$,        ▷ Pseudo-inverse
4: $\hat{\mathbf{z}}, \cdot = \mathcal{E}(\mathbf{y}); \mathbf{z}, \mathbf{r} = \mathcal{E}(\mathbf{A}^\dagger\mathbf{y})$,        ▷ Encoder
5: $\epsilon = \mathbf{z} - f_\phi(\hat{\mathbf{z}}, c_{\text{null}}, \tau_i)$,        ▷ Fix range-space content
6: $\Delta\epsilon = (1 + w_1) f_\phi(\hat{\mathbf{z}}, c_n, \tau_i) - (1 + w_2) f_\phi(\hat{\mathbf{z}}, c_l, \tau_i) + (w_2 - w_1) f_\phi(\hat{\mathbf{z}}, c_{\text{null}}, \tau_i)$,        ▷ Refine null-space content
7: $\bar{\mathbf{z}} = f_\phi(\hat{\mathbf{z}}, c_{\text{null}}, \tau_i) + \epsilon + \left(\mathbf{I} - \gamma\mathbf{A}^\dagger\mathbf{A}\right)\Delta\epsilon \to \bar{\mathbf{z}} = \mathbf{z} + \left(\mathbf{I} - \gamma\mathbf{A}^\dagger\mathbf{A}\right)\Delta\epsilon$,    ▷ Predict result
8: **for** $\tau_i = T - 1, T - 2, \ldots, \kappa$ **do**        ▷ Iterative refinement
9:      $\hat{\mathbf{z}} \to \hat{\mathbf{z}}_{\tau_i}$
10:      $\epsilon = \tilde{\mathbf{z}} - f_\phi(\hat{\mathbf{z}}_{\tau_i}, c_{\text{null}}, \tau_i)$
11:      $\Delta\epsilon = (1 + w_1) f_\phi(\hat{\mathbf{z}}_{\tau_i}, c_n, \tau_i) - (1 + w_2) f_\phi(\hat{\mathbf{z}}_{\tau_i}, c_l, \tau_i) + (w_2 - w_1) f_\phi(\hat{\mathbf{z}}_{\tau_i}, c_{\text{null}}, \tau_i)$,
12:      $\bar{\mathbf{z}} = \mathbf{z} + \left(\mathbf{I} - \gamma\mathbf{A}^\dagger\mathbf{A}\right)(\epsilon + \Delta\epsilon)$
13:      $\tilde{\mathbf{z}} = \bar{\mathbf{z}}$
14: **end for**
15: **Return** $\bar{\mathbf{x}} = \mathcal{D}(\bar{\mathbf{z}}, \mathbf{r})$.        ▷ Decoder

---

content and textures from low-light images, ensuring consistency is maintained. Our objective is to implement language-aware illumination attention that enhances illumination while preserving the original content of the low-light image. Specifically, we use the original $\mathbf{Q}_n$, $\mathbf{K}_n$, and $\mathbf{V}_n$ in self-attention mechanism. Subsequently, we query semantically similar content from $\mathbf{K}_l$ and $\mathbf{V}_l$ using the target query $\mathbf{Q}_n$. The attention mechanism can be expressed in matrix form as follows:

$$\text{Attention}(\mathbf{Q}_n, \mathbf{K}_l, \mathbf{V}_l) = \text{Softmax}\left(\frac{\mathbf{Q}_n\mathbf{K}_l^T}{\sqrt{d}}\right)\mathbf{V}_l. \tag{10}$$

**Lighting Decoder.** Once the range-null content in the latent space is refined, we propose a lighting decoder that converts $\bar{\mathbf{z}}$ to image space as $\bar{\mathbf{x}}$. This entire process can be expressed as follows:

$$\mathbf{z}, \mathbf{r} = \mathcal{E}(\mathbf{x}), \tag{11}$$

$$\bar{\mathbf{x}} = \mathcal{D}(\bar{\mathbf{z}}, \mathbf{r}). \tag{12}$$

where $\bar{\mathbf{z}}$ is refined from $\mathbf{z}$ in equation 9, and $\mathbf{r}$ is the middle-layer feature. Our lighting decoder incorporates additional convolutional layers for hidden feature fusion and utilizes skip connections. To our knowledge, QuadPrior Wang et al. (2024) introduced a bypass decoder that effectively captures details from randomly degraded images and is sensitive to illumination changes. However, it has notable limitations: the bypass decoder is trained on the COCO dataset Lin et al. (2014), which contains underexposed and low-quality images, thereby compromising its decoding capabilities. Additionally, its performance in recovering complex textures under varying illumination conditions is constrained, as the random illumination jittering and noise do not integrate a physical model of LLIE.

To address the aforementioned issues, we collect a well-exposed image dataset from benchmark sources and the internet to ensure high-quality training data. We then fine-tune the lighting decoder to accept both Retinex-driven degraded images and normal images, allowing it to adapt to illumination degradation effectively. This approach ensures that the model remains sensitive to variations in illumination within the latent space while preserving high fidelity through skip connections. The overall objective for training the decoder model can be formulated as follows:

$$L = \min\max\left(L_{\text{rec}}\left(\mathbf{x}, \mathcal{D}\left(\mathcal{E}(\tilde{\mathbf{x}}), \mathbf{r}\right)\right) + L_{\text{reg}}\left(\mathbf{z}; \mathcal{E}, \mathcal{D}\right)\right) \tag{13}$$

where $\tilde{\mathbf{x}}$ is the Retinex-driven degraded images. $L_{\text{rec}}$ denotes the pixel-wise reconstruction losses, including MSE and LPIPS, while $L_{\text{reg}}$ regularizes the latent $\mathbf{z}$ to be zero centered and small variance.

It is worth noting that our method also supports rapid one-step generation while also enabling multi-step sampling, allowing for a trade-off between computational efficiency and enhancement quality, much like traditional consistency models.

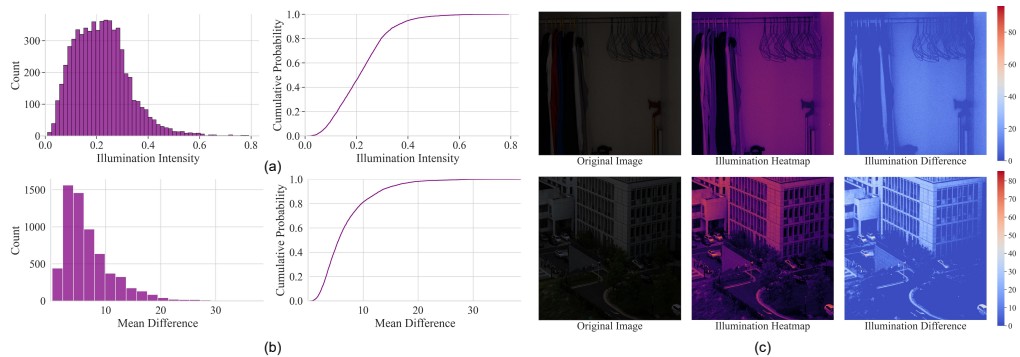

Figure 4: The distribution and intensity of average and difference illumination from the statistics of the low/normal light image pairs.

### 2.3 CONSTRUCTING ILLUMINATION FACTOR **A**

Rather than focusing on precise illumination estimation, we reconsider Retinex theory through the lens of Range-null space decomposition and propose a simple yet effective global illumination intensity. We introduce the following proposition:

**Proposition 1.** *We assume that the low-light image mainly suffers from the global degeneration operator $\varpi$ in illumination intensity* **A***, which is a non-zero constant matrix. When we set the constant of* **A** *as $\varpi$, its pseudo-inverse* $\mathbf{A}^\dagger$ *can be easily calculated as*

$$\mathbf{A}^\dagger = \frac{1}{\varpi}\mathbf{I}. \tag{14}$$

Hence, we present a simple yet interpretable range-space content $\mathbf{A}^\dagger\mathbf{y}$ for the LLIE task. It is then encoded into latent space as $\mathbf{z} = \mathcal{E}(\mathbf{A}^\dagger\mathbf{y})$. The proof can be found in the appendix B.

To validate the universality of global illumination reduction, we gather approximately 7,000 low and normal light image pairs from existing benchmark datasets, including LOLv1 Wei et al. (2018), LOLv2 Yang et al. (2021), LSRW Hai et al. (2023), and our UAV-LL dataset. We conducted a study on the illumination degradation between these paired images. Specifically, we first estimated the average global illumination intensity. Following the methodology of Guo et al. (2017), we computed pixel-wise illumination by selecting the maximum value across the three color channels as the global illumination and then calculated the difference between this value and the average illumination.

As illustrated in Figure 4(a), the global illumination intensity in low-light images is low and uniformly distributed between 0.1 and 0.4. Additionally, Figure 4(b) shows that the average difference intensity predominantly ranges from 0 to 15, indicating that the pixel-wise illumination is close to the average illumination. By observing toy examples of the illumination and difference maps in Figure 4(c), we have a reason to believe that most low-light images exhibit characteristics of global degradation.

## 3 EXPERIMENTS

### 3.1 DATASETS AND IMPLEMENTATION DETAILS

**Datasets.** We evaluate our method on two paired benchmark datasets: LOL+ and LSRW Hai et al. (2023). Following Wang et al. (2024), we adopt LOL+ consists of 115 low and normal light image pairs from LOLv1 Wei et al. (2018) and LOLv2 Yang et al. (2021), while LSRW contains 50 pairs. To further assess generalization capabilities, we introduce a new UAV-LL dataset, referred to Appendix C, comprising 300 image pairs captured in diverse mobile environments For fine-tuning the lighting decoder, we gathered 7,000 normal images from benchmark datasets and the internet.

**Compared Methods.** We compare our method with five SOTA supervised methods, including URetinex-Net Wu et al. (2022), R2RNet Hai et al. (2023), Retinexformer Cai et al. (2023), GSAD Hou et al. (2023) and DiffLL Jiang et al. (2023), all of which achieve state-of-the-art results on benchmark datasets. Furthermore, we compare our method with nine unsupervised low-light image

Table 1: Quantitative comparisons on LOL+, LSRW and UAV-LL dataset. The best unsupervised result is in red color, while the second best result is in blue color under the unsupervised setting.

| | Method | LOL+ | | | LSRW | | | UAV-LL | | | Training Set |
|---|---|---|---|---|---|---|---|---|---|---|---|
| | | PSNR | SSIM | LPIPS | PSNR | SSIM | LPIPS | PSNR | SSIM | LPIPS | |
| SL | URetinex-Net(CVPR22) | 20.93 | 0.854 | 0.104 | 18.27 | 0.518 | 0.295 | 13.35 | 0.482 | 0.276 | LOL |
| | R2RNet (JVCIR23) | 20.20 | 0.816 | 0.664 | 16.24 | 0.502 | 0.251 | 11.61 | 0.49 | 0.295 | LSRW |
| | Retinexformer(ICCV23) | 23.98 | 0.843 | 0.117 | 15.98 | 0.467 | 0.242 | 12.10 | 0.624 | 0.207 | LOL |
| | GSAD(NeurIPS23) | 28.32 | 0.886 | 0.093 | 17.41 | 0.507 | 0.294 | 14.75 | 0.521 | 0.281 | LOL |
| | DiffLL(Siggraph Asia23) | 28.54 | 0.870 | 0.102 | 13.48 | 0.396 | 0.264 | 11.69 | 0.595 | 0.247 | LOL |
| USL | EnlightenGAN(TIP21) | 18.57 | 0.700 | 0.302 | 17.10 | 0.462 | 0.322 | 17.96 | 0.538 | 0.252 | own data |
| | ZeroDCE(CVPR20) | 17.64 | 0.572 | 0.316 | 15.86 | 0.443 | 0.315 | 17.91 | 0.502 | 0.293 | own data |
| | SCI(CVPR22) | 16.97 | 0.532 | 0.312 | 15.24 | 0.419 | 0.321 | 14.78 | 0.576 | 0.240 | LOL+ |
| | PairLIE(CVPR23) | 19.70 | 0.774 | 0.235 | 17.60 | 0.501 | 0.323 | 15.21 | 0.512 | 0.248 | LOL+ |
| | NeRCo(ICCV23) | 19.67 | 0.720 | 0.266 | 17.84 | 0.535 | 0.371 | 16.24 | 0.530 | 0.459 | LSRW |
| | CLIP-LIT(ICCV23) | 14.82 | 0.524 | 0.371 | 13.48 | 0.396 | 0.264 | 17.22 | 0.558 | 0.282 | own data |
| | Zero-IG(CVPR24) | 22.17 | 0.771 | 0.276 | 16.61 | 0.470 | 0.282 | 12.66 | 0.362 | 0.377 | LOL |
| | QuadPrior(CVPR24) | 20.31 | 0.808 | 0.202 | 17.17 | 0.558 | 0.199 | 19.51 | 0.674 | 0.272 | COCO |
| | LightenDiff(ECCV24) | 20.45 | 0.803 | 0.192 | 18.55 | 0.539 | 0.311 | 16.88 | 0.547 | 0.249 | LOL |
| | **Our RLPCM** | 24.07 | 0.837 | 0.105 | 19.11 | 0.570 | 0.190 | 21.17 | 0.613 | 0.266 | Normal |

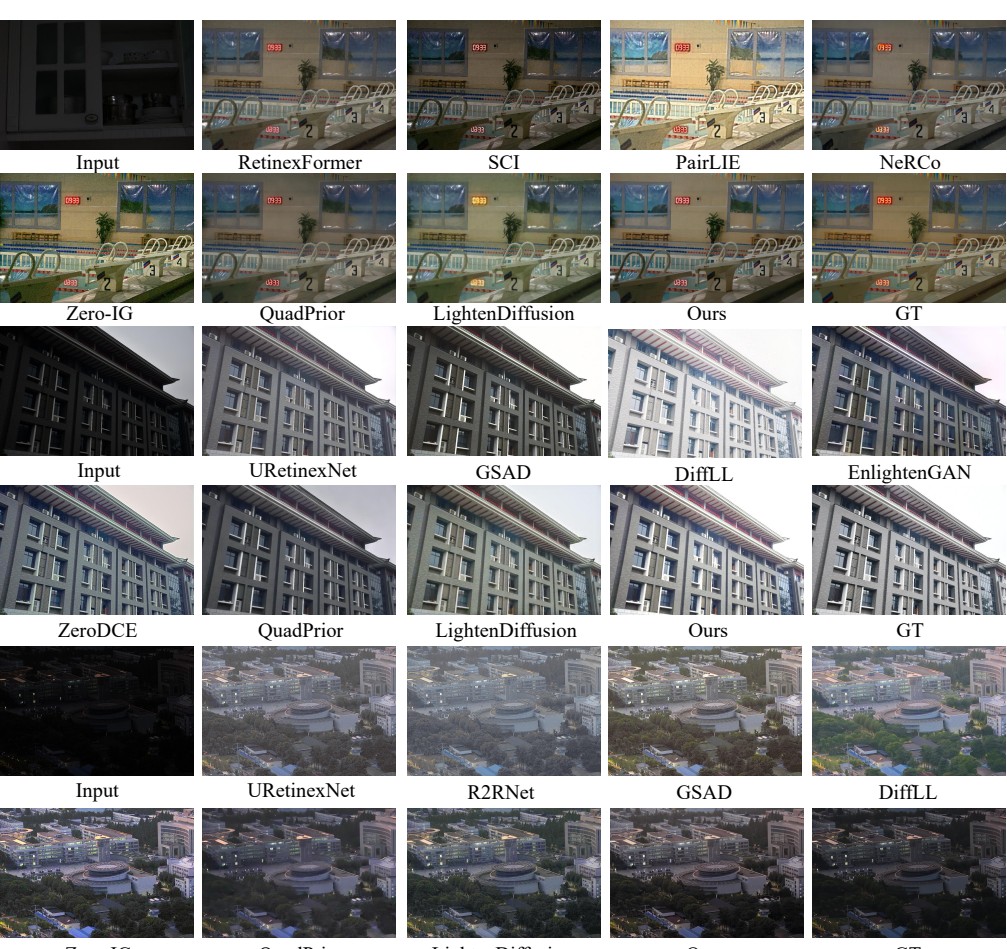

Figure 5: Qualitative comparison with previous methods on LOL+, LSRW and UAV-LL datasets. Our RLPCM effectively improve the realness and preserves the details compared to other methods.

enhancement approaches, including EnlightenGAN Jiang et al. (2021), ZeroDCE Guo et al. (2020), SCI Ma et al. (2022), PairLIE Fu et al. (2023), NeRCo Yang et al. (2023), CLIP-LIT Liang et al. (2023), QuadPrior Wang et al. (2024), and LightenDiffusion Jiang et al. (2024a). Additionally, we evaluate the cross-dataset generalization of these pre-trained models by applying to UAV-LL datasets.

**Implementation Details.** We implement RLPCM in PyTorch Paszke et al. (2019) on a server with the 4090GPUs. In our framework, LCM is tuning-free, only needs to fine-tune the lighting decoder.

Figure 6: Qualitative comparison with diffusion-based image restoration Methods on LOLv1 datasets. Our method yields the most reasonable and satisfactory results across all methods +.

Table 3: The ablation study on LOL+ and LSRW datasets. The best result is in red color.

| Method | | LOL | | | LSRW | | |
|---|---|---|---|---|---|---|---|
| | | PSNR | SSIM | LPIPS | PSNR | SSIM | LPIPS |
| Range-null Space Refinment | w/o Range-space content | 22.90 | 0.805 | 0.128 | 18.92 | 0.557 | 0.239 |
| | w/o Null-space content | 15.01 | 0.555 | 0.169 | 14.06 | 0.394 | 0.270 |
| Decoder | vanilla decoder | 22.04 | 0.729 | 0.131 | 18.20 | 0.529 | 0.229 |
| | bypass decoder | 23.01 | 0.787 | 0.148 | 19.08 | 0.545 | 0.267 |
| Ours | | 24.07 | 0.837 | 0.105 | 19.11 | 0.570 | 0.204 |

We set the batch size to 8 and train for 140k steps, with an initial learning rate of 1e-4 using the ADAM optimizer. For evaluation, we report peak signal-to-noise ratio (PSNR), structural similarity (SSIM) and LPIPS Zhang et al. (2018) is selected as the evaluation metrics.

## 3.2 MAIN RESULTS

**Quantitative Comparison.** As shown in Table 1, we evaluated the performance of our RLPCM with five SOTA supervised methods and nine unsupervised methods. Our RLPCM surpasses all unsupervised methods on LOL dataset in terms of PSNR SSIM and LPIPS, and achieves comparable performance with the supervised methods, while our approach achieves robust performance across all datasets, further emphasizing its generalizability and effectiveness.

To further validate the performance of existing methods, we use the released models trained using their own data for evaluation on our proposed UAV-LL dataset. Despite the current diffusion models, GSAD and DiffLL, achieve the SOTA results in LOL datasets, ones exhibit the limited performance to new scenarios. In contrast, the unsupervised diffusion methods ,i.e., our model, QuadPrior and LightenDiffusion, outperform than all pre-trained supervised methods, and our approach has the better result than others.

**Qualitative Comparison.** Figure 5 summarises the vision results of our method with other SOTA methods among LOL+, LSRW and UAV-LL datasets. It is observed that existing methods suffer from overexposure or underexposure illumination and noise, while our method provide a proper illumination and effectively suppress the noise. Notably, the ground-truth image of UAV-LL dataset has a trade-off between illumination and noise, since our UAV-LL is captured on the real low light scenarios. The limitation and more results and are provided in Appendixes D and E, respectively.

## 3.3 COMPARING WITH DIFFUSION-BASED IMAGE RESTORATION METHODS

To further verify the effectiveness of our framework, we compare our approach with several state-of-the-art (SOTA) diffusion-based image restoration methods, including GDP Ben Fei (2023), DDNM Wang et al. (2023b), and AutoDIR Jiang et al. (2024b). Following the evaluation method used in GDP, we assess the results on the LOLv1 dataset Wei et al. (2018). As shown in Table 2, our method achieves the best perfor-

Table 2: Quantitative comparison with diffusion-based image restoration methods on LOLv1 datasets. The best result is in red color.

| Method | PSNR | SSIM | LPIPS | Time (S) |
|---|---|---|---|---|
| GDP (CVPR 23) | 13.93 | 0.630 | 0.680 | 60.00+ |
| DDNM (ICLR 23) | 13.15 | 0.492 | 0.498 | 5.47 |
| AutoIR (ECCV 24) | 19.95 | 0.811 | 0.107 | 28.98 |
| Ours | 24.12 | 0.826 | 0.103 | 0.84 |

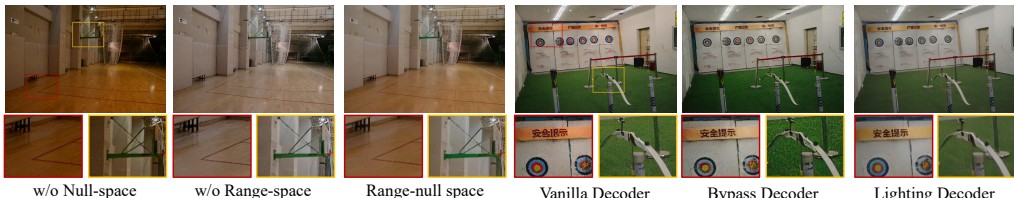

Figure 7: Visual results of the ablation study on the Range-null space and Decoder, our proposed full model exhibits improved detail handling in the local images.

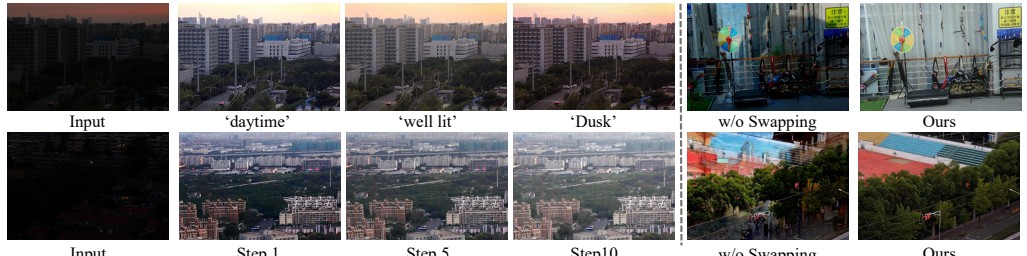

Figure 8: Visual results of the ablation study on natural language guidance, iteration steps, and the self-attention swapping mechanism.

mance on the LOLv1 dataset, while our inference speed is significantly higher than that of other models. This improvement is attributed to the introduction of a consistency model, which reduces the number of iterations. Furthermore, our latent prior-guided framework does not reduce the inference efficiency. Besides, Due to the limitation of DDNM, i.e, explicit degradation priors, we introduce our proposed global degeneration operator into DDNM, which also yields an acceptable results. In Figure 6, our method yields the most reasonable and satisfactory results across all methods.

### 3.4 ABLATION STUDY

We present ablation studies to demonstrate the effectiveness of the each part in our proposed RNLP. For range-null space, we remove the range-space content and null-space content, respectively. For refining the null-space content in latent space, we remove the self-attention swapping mechanism, while discuss the effect of different language prompts and its guidance scale. For the fidelity, we compare the lighting decoder with vanilla decoder, consistency decoder and bypass decoder.

The quantitative restuls of the ablated study on LOL+ and LSRW are presented in Table 3. Overall, the range-null space refinement provide a key contribute to achieve the best performance of the full model. Without either the range-space or null-space content, there would be a rapid decline in performance. In comparison, the impact of the decoder on the results is relatively smaller. As shown in Figure 7 a suboptimal decoder may lead to color distortion or content degradation. Furthermore, we compare the influence of natural language guidance, iteration steps, and the self-attention swapping mechanism in Figure 8. Specifically, natural language guidance can appropriately alter the tone of an image without impacting its content. However, in the absence of the self-attention swapping mechanism, irrelevant content may be introduced, highlighting the necessity of this mechanism. Lastly, the results indicate that our method requires only a few steps (or even just one) to achieve relatively satisfactory results.

## 4 CONCLUSION

In this paper, we presented a range-null latent prior-guided consistency model, a novel approach that introduces an off-the-shelf consistency model into low-light image enhancement using Retinex-based range-null space decomposition. Additionally, we contributed a new UAV LLIE dataset for comprehensive evaluation. Extensive experiments on both benchmark and UAV-LL datasets demonstrate that our model achieves robust performance. In future work, we aim to explore model distillation and extend the latent prior-guided framework to low-light video enhancement tasks.

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

# A  RELATED WORK

**Low Light Image Enhancement**.

low-light image enhancement Lore et al. (2017) is a critical task, influencing both human visual perception and the related computer vision applications such as depth estimation Wang et al. (2021), object detection Hashmi et al. (2023), semantic segmentation Pan et al. (2024).

To our knowledge, LLNetLore et al. (2017) is a pioneer, taking the lead in introducing deep neural networks into the field of low-light image enhancement, achieving remarkable results through supervised learning. Subsequently, LightenNetCai et al. (2018) used a convolutional neural network (CNN) to attempt contrast enhancement for a single image. MBLLENLv et al. (2018) further innovates and introduces a multi-branch fusion strategy within the CNN architecture to capture and fuse richer image features. In addition, a series of SOTA methods such as SNR-Net Xu et al. (2022), Retinexformer Cai et al. (2023), DiffLL Jiang et al. (2023), and MambaLLIE Weng et al. (2024), have attracted considerable attention due to their impressive performance on various low-light enhancement benchmark datasets (e.g., MIT Bychkovsky et al. (2011), LOL Wei et al. (2018), LSRW Hai et al. (2023)). However, despite leveraging advanced network designs incorporating Retinex theory Land & McCann (1971) , Transformers Vaswani et al. (2017), state-space models Gu & Dao (2024), and diffusion models Ho et al. (2020), these supervised methods exhibit limited generalization to unseen scenarios. This is likely due to the comparatively small training datasets, which fail to capture diverse illumination conditions and device degradations. Hence, reducing the reliance on paired image collections for low-light enhancement remains a significant challenge, particularly in mobile environments.

To address this, recent unsupervised approaches have focused on leveraging unpaired datasets or even single low/normal light images datasets for training, thereby reducing the dependency on paired images. Prior state-of-the-art methods, ZeroDCEGuo et al. (2020), RUASLiu et al. (2020) and their subsequent studies such as Ma et al. (2022), Fu et al. (2023), Wang et al. (2024), etc., use physical lighting priors as guidance to achieve image enhancement without external supervision. Currently, diffusion-based unsupervised models Wang et al. (2024); Jiang et al. (2024a) have attracted significant attention for their powerful generative capabilities. QuadPrior Wang et al. (2024) introduces an illumination-invariant prior as a conditional generative model using the ControlNet-shape framework, where the entire architecture is trained on COCO dataset Lin et al. (2014). LightenDiffusion employs a diffusion-based model, utilizing unpaired images by decomposing them into reflectance and illumination maps, which serve as latent space inputs for low-light enhancement.

**Diffusion Model**. In recent years, diffusion models have been widely used in image generation tasks. At the same time, significant progress has been achieved in low-level vision tasks. RePaint Lugmayr et al. (2022) utilize pre-trained DDPM as a generative prior to generate high-quality, diverse inpainted images without the need for mask-specific training. For low-level vision task, SR3 Saharia et al. (2021) adopted a conditional image generation method based on a noise diffusion probability model to achieve image super-resolution through iterative refinement, LPDM Panagiotou & Bosman (2023) introduced the Low-light Post-processing Diffusion Model (LPDM) to model the conditional distribution between low-light images and normal exposure images. DiffPIR Zhu et al. (2023) combined the traditional interpolation image restoration method with a diffusion sampling framework, aiming to exploit the diffusion model as a prior for a generative denoiser. DiffLL Jiang et al. (2023) proposed a wavelet conditional diffusion model (WCDM) that combines the advantages of wavelet transform and the generation capability of diffusion model to achieve high-quality image enhancement. Diff-Retinex Yi et al. (2023) rethink the low-light image enhancement task by combining a physically interpretable model and a generative diffusion model. LatentFD Mei et al. (2023) utilized a latent feature-guided diffusion model to achieve efficient shadow removal. JoReS-Diff Wu et al. (2023) improved the generation ability of the diffusion model by introducing Retinex theory as an additional preprocessing condition. GASD Hou et al. (2023) proposed a global structure-aware diffusion process for low-light image enhancement through global structure awareness and uncertainty-guided regularization. PA-Diff Zhao et al. (2024) proposed a new UIE framework that aims to utilize physical knowledge to guide the diffusion process. MDMS Shang et al. (2024) enables the model to adaptively learn the noise distribution and thus improve the quality of the generated image by introducing a space-frequency domain fusion module and combining a multi-domain learning paradigm and a multi-scale sampling strategy. CFWD Xue et al. (2024) proposed a wavelet diffusion model based on

CLIP and Fourier transform guidance, which uses multi-modal visual language information in the frequency domain space generated by multiple wavelet transforms to guide the enhancement process.

## B  SOLVING PSEUDO-INVERSE $\mathbf{A}^{\dagger}$

Consider an image $\mathbf{x}$ of size $m \times n$. We can vectorize the image $\mathbf{x}$ into a column vector $\vec{x} \in \mathbb{R}^{mn \times 1}$. Let $\mathbf{A} \in \mathbb{R}^{mn \times mn}$ be a degradation operator with a $1 \times 1$ global degradation operator $\varpi$. This matrix can be written as:

$$\mathbf{A} = \begin{pmatrix} \varpi & 0 & \cdots & 0 \\ 0 & \varpi & \cdots & 0 \\ \vdots & \vdots & \ddots & \vdots \\ 0 & 0 & \cdots & \varpi \end{pmatrix}_{mn \times mn} \tag{15}$$

Thus, the degradation operator $\mathbf{A}$ can be represented as $\mathbf{A} = \varpi \cdot \mathbf{I}_{mn}$.

The pseudo-inverse $\mathbf{A}^{+}$ of a matrix $\mathbf{A}$ is defined as the matrix that satisfies the following conditions:

$$\mathbf{A}\mathbf{A}^{\dagger}\mathbf{A} = \mathbf{A} \tag{16}$$

Substituting back,

$$\varpi \cdot \mathbf{I}_{mn}\mathbf{A}^{\dagger}\varpi \cdot \mathbf{I}_{mn} = \mathbf{A}. \tag{17}$$

Thus, the generalized inverse $\mathbf{A}^{\dagger}$ is

$$\mathbf{A}^{\dagger} = \frac{1}{\varpi^2}\mathbf{A} = \frac{1}{\varpi}\mathbf{I}_{mn} \tag{18}$$

## C  DETAILS OF UAV-LL DATASET

The UAV-LL dataset used in our experiments primarily consists of drone-view urban scenes, captured with a 4/3 CMOS Hasselblad camera in real-world settings during dusk and nighttime conditions. The UAV-LL dataset contains 300 pairs of drone-view data, including a large variety of scenes with various real noises, and different darkness levels.

To obtain authentic low-light images, we meticulously adjusted exposure, ISO, and other parameters to capture ground truth (GT) images in genuine low-light environments. This process often entails a trade-off between visibility and image noise. Some samples from our UAV-LL dataset are displayed in Figure 9.

## D  LIMITATIONS

Our approach remain many limitations that deserve further study:

1) Our proposed global illumination operator, while simple and efficient, may struggle in high-contrast scenes. One potential solution is to utilize null-space decomposition for content refinement.

2) Despite employing LCM, our method still faces challenges in directly handling high-resolution images, primarily due to the constraints of the pre-trained model and limited computational resources.

As shown in Figure 10, Zero-IG and ours full model are unable to obtain proper illumination, resulting in the overexposure results. In contrast, our method without Range-space content yields a better enhanced result.

## E  MORE VISUAL RESULTS

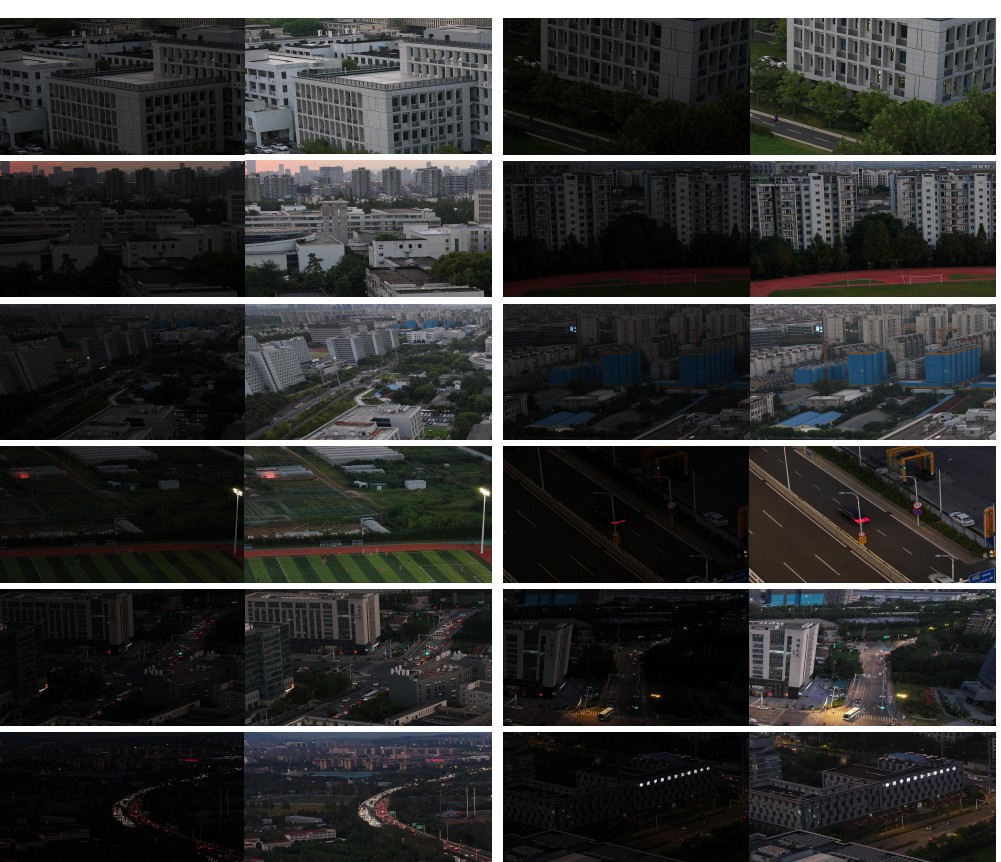

Figure 9: Samples of the UAV-LL dataset.

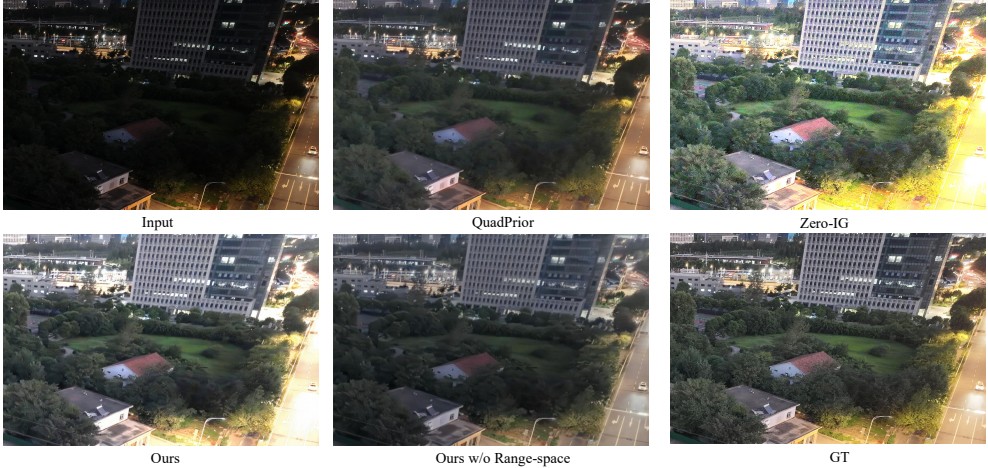

Figure 10: Visual comparison of the unsupervised methods on high-contrast scenes.

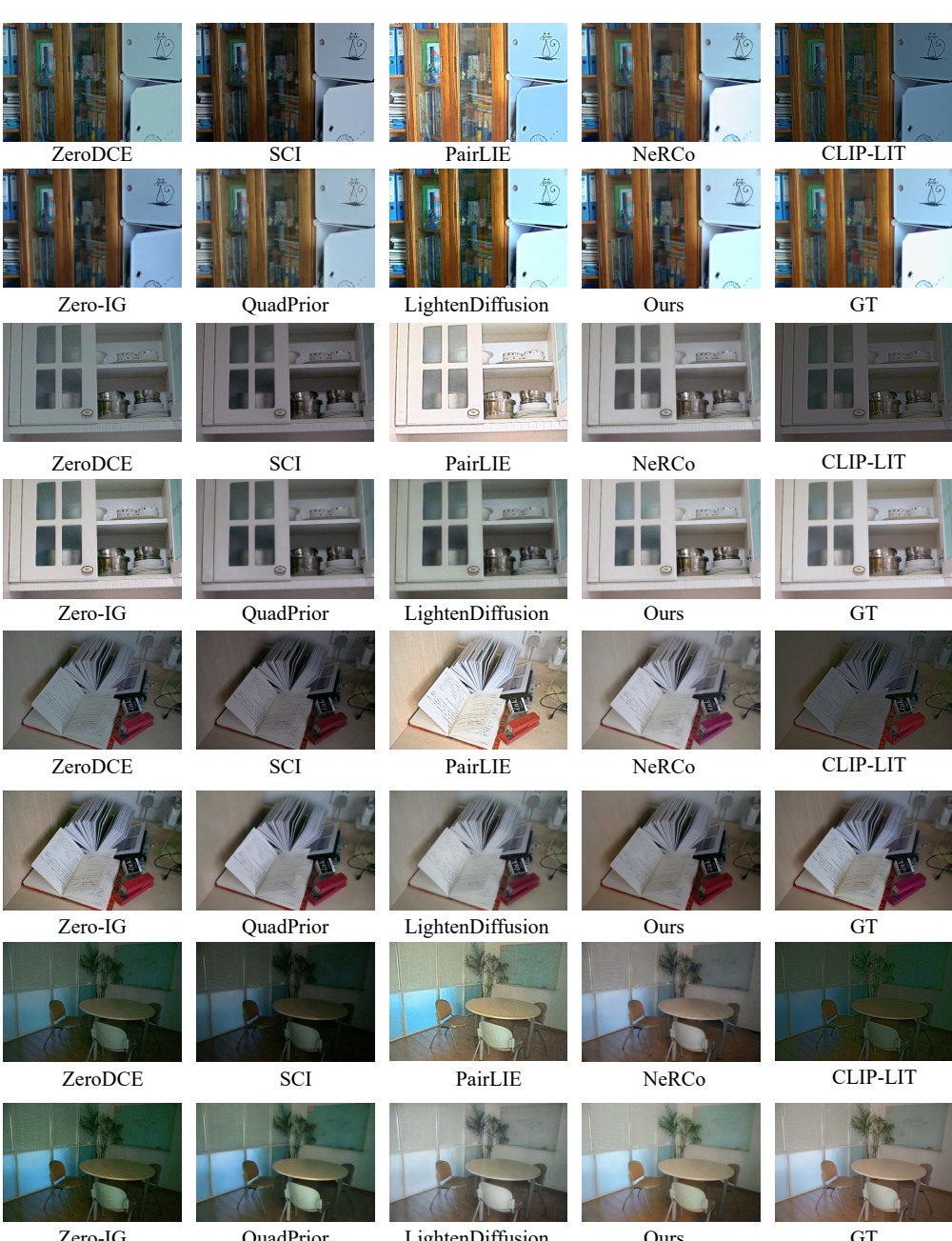

Figure 11: Visual comparison of the SOTA unsupervised methods on LOL+ datasets.

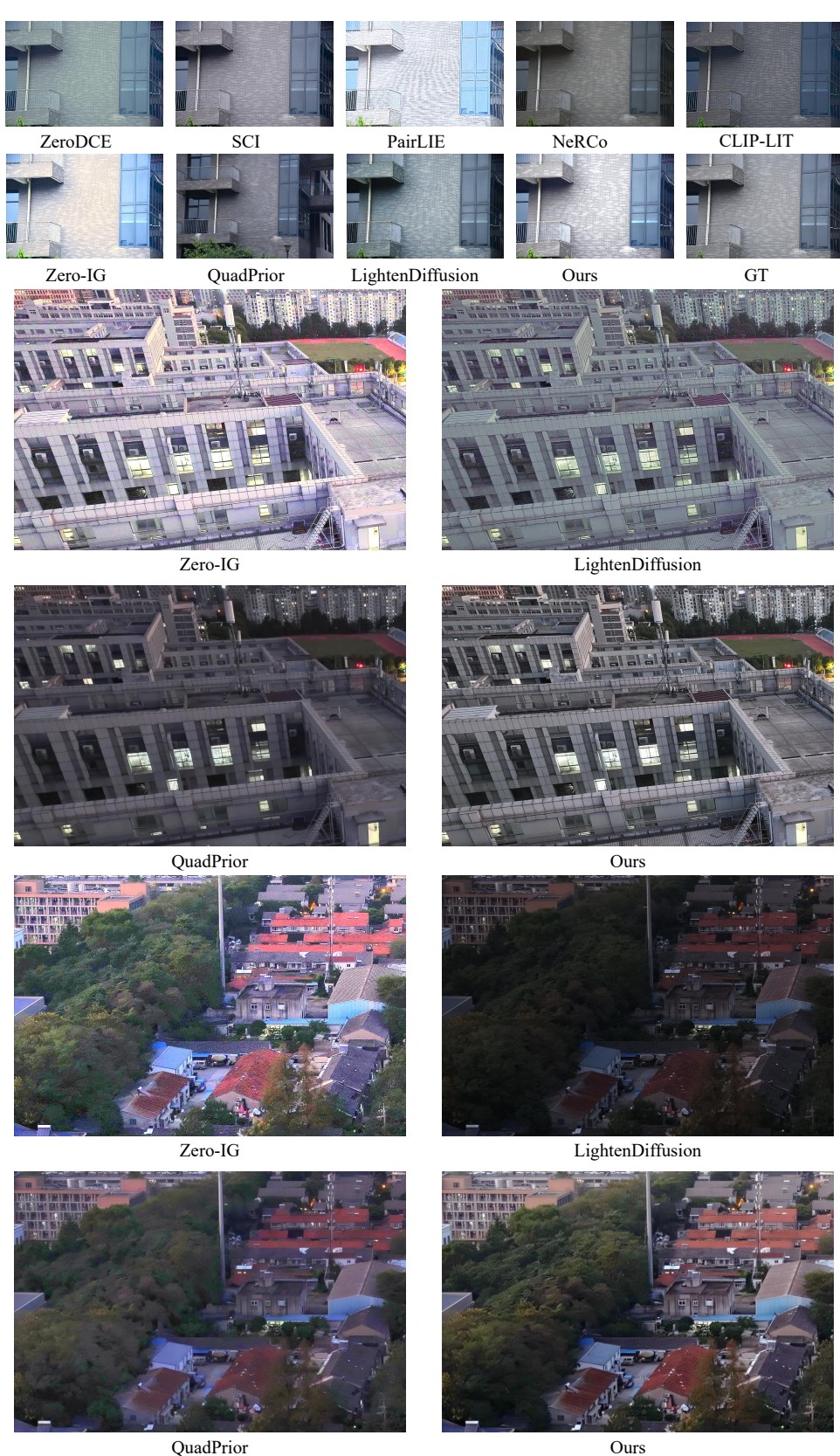

Figure 12: Visual comparison of the SOTA unsupervised methods on LSRW and UVA-LL datasets.

