# OpenReview forum: "Range-Null Latent Prior-guided Consistency Model for Low Light Image Enhancement"
_ICLR.cc/2025/Conference — ICLR 2025 Conference Withdrawn Submission_

### Official Review · Reviewer_bn7U · 2024-10-17

**Soundness:** 1
**Presentation:** 3
**Contribution:** 2
**Rating:** 1
**Confidence:** 4

**Summary:**

This manuscript presents a new low-light enhancement model with range-null space latent prior guidance. Classifier-free guidance from the language model is fed into the model to improve the quality of the enhanced images. The authors also proposed a new evaluation dataset with low-light images captured with UAV. Experiments demonstrate the effectiveness of the proposed method.

**Strengths:**

+ Decent performance on benchmark datasets.
+ Clear, easy-to-follow manuscript.

**Weaknesses:**

+ The null space, say, {x for Ax=0} of a full-rank linear operator A (as the author formulated in Eq.15) only contains x=0. The null-space refinement term $I - \gamma A^{-1}A$ thereby degrades to $1 - \gamma$. Thus the claimed null-space refinement is over-claim.
+ The experiments contain unfair comparisons. The light decoder is trained with benchmark images as the author stated in the implementation details. This renders an unfair comparison for all competitors and raises doubts about the proposed method's soundness. Authors shall use COCO, HQ-50k, or other normal-light datasets that are out of the benchmarking dataset for a fair comparison.
+ The dataset proposed is not well-motivated. The authors shall highlight the contribution to the community for this evaluation set.
+ Please proofread to eliminate typos.
+ I suggest the authors to move the related work part to the main manuscript, since this year the page limit of the conference is 10 pages.

**Questions:**

+ How could this be "null-space refinement" since there is just a single point (x=0) inside the null space?
+ The ablation studies are not comprehensive. What if we use vanilla CFG?
+ Numerous datasets exist for low-light enhancement benchmarking. Why do we need another evaluation dataset? What is the advantage of the proposed dataset? This part is not well-motivated.
+ The light decoder is trained with benchmark images as the author stated in the implementation details. This renders an unfair comparison for all competitors and raises doubts about the proposed method's soundness. Authors shall use COCO, HQ-50k, or other normal-light datasets that are out of the benchmarking dataset for a fair comparison.

---

### Official Review · Reviewer_zvFr · 2024-10-20

**Soundness:** 3
**Presentation:** 2
**Contribution:** 2
**Rating:** 3
**Confidence:** 4

**Summary:**

This paper proposes a latent consistency model (LCM, (Luo et al., 2023))-based low-light image enhancement method, called range-null latent prior-guided consistency model (RLPCM). RLPCM first obtains an illumination degradation factor and uses the degradation factor to compute the range-space content, which is then transformed into the latent space via the off-the-shelf LCM (Luo et al., 2023). RLPCM then uses language prompts to refine the null-space content and uses a swapping self-attention mechanism (Catet al., 2023; Xu et al. 2024) for enhancing consistency. The paper also collects a UAV dataset consisting of 300 low-/normal-light image pairs.

The paper tries to address an existing task with a new approach. ***I am on the fence but leaning to reject this paper due to the insignificant idea and novelty, and the insufficient results.***

**Strengths:**

+ It is good that the paper explores newly developed techniques (e.g., consistency models, range-null space decomposition, and language prompts) for enhancing low-light images.

+ The quantitative results of RLPCM in Table 1 are impressive.

**Weaknesses:**

***Insignificant Main Idea.***
Given that the range-null space decomposition (Schwab et al., 2019) has been explored with the diffusion model in (Wang et al., 2023b) for image restoration tasks, and consistency models (Song et al., 2023) have offered quality-computation trade-off against diffusion models via direct noise-to-data mapping, using latent consistency model (Luo et al., 2023) to replace the diffusion model in (Wang et al., 2023b) and applies it for a specific task of enhancing low-light images is incremental. The authors may highlight which part of the proposed range-null latent prior-guided framework is novel and justify its significance via analysis.

***Unclear Motivation.***
In addition, the logic of introducing language prompts for refining null-space content is hard to follow. First, I cannot find where in CFG (Ho&Salimans, 2022) mentions about generating language-aligned images. Second, it is not clear how the language-aligned image generation inspires this paper to use contrasted language prompts to refine null-space and how these contrasted language prompts are obtained. The authors may explicitly explain the motivation for using contrasted language prompts for refining the null-space content of low-light images.

***Unclear Novelty of the Proposed Lighting Decoder.***
The differences between the proposed lighting decoder and the bypass decoder of QuadPrior (Wang et al., 2024) are the different training data. The authors may clarify whether it is novel and (if yes) justify how it represents significant novelty.

***Unsupported Claims.***
The paper claims that the global illumination prior “physically” guarantees the reliability of range-space content. The paper also claims that the consistency model (Song et al., 2023) is more effective for generating null-space content. While these two claims are important to understanding the model designs,  there is no evidence and analysis that can justify these two claims. The authors may provide analysis and evidence to justify them.

***Unconvincing comparisons.***
The ground truth images in the proposed UAV-LL datasets are still quite dark, for example, the last case in Figure 5, and the result produced by the proposed method is also darker than those of other methods. While the paper criticizes previous methods for not generalizing well to the proposed dataset according to the quantitative comparisons in Table 1, it seems that existing methods produce visually better results but the ground truth images in the proposed dataset favor the results produced by this paper. This makes the comparisons on the proposed dataset unconvincing. The authors may clarify why the evaluation based on such dark ground truth images in the UAV-LL dataset is reasonable.

***Missing ablation studies.***
The results of removing the self-attention swapping mechanism and using different language prompts and guidance scales are missing in Table 3.

***Minor Issues.***
Line 070, Should the reference for LCM be (Luo et al., 2023) instead of (Song et al., 2023) for the consistency models?
Line, 086, this -> This.
Line, 398-399, we report…is selected as…
Line, 462, should RNLP be RLPCM?
Line 377, it is not clear how many GPUs are used.

It is not clear whether the authors would release the training data and codes. Without releasing the training data, it may not be possible to reproduce the results as the paper relies on self-collected training data.

**Questions:**

1. The authors may highlight which part of the proposed range-null latent prior-guided framework is novel and justify its significance via analysis.

2. The authors may explicitly explain the motivation for using contrasted language prompts for refining the null-space content of low-light images.

3. The authors may clarify whether the lighting decoder is novel and (if yes) justify how it contains significant novelty.

4. The authors may provide analysis and evidence to justify the two claims of the global illumination prior "physically generating the reliability of range-space content" and the consistency model is "more effective to generate null-space content".

5. The authors may clarify why the evaluation based on such dark ground truth images in the UAV-LL dataset can be reasonable.

6. Please report the quantitative results of removing the self-attention swapping mechanism and using different language prompts.

---

### Official Review · Reviewer_dVZb · 2024-10-31

**Soundness:** 3
**Presentation:** 3
**Contribution:** 3
**Rating:** 6
**Confidence:** 4

**Summary:**

This paper presents RLPCM, a unsupervised model for low light image enhancement, which incoporates range-null space decomposition with retinex theory into consistency model, with a range-null latent prior-duided framework. Experiments are evaluated on the leading benchmarks with comparisons to the recent LLIE methods, showing improvements in evaluated numbers. Ablation studies are privided to validate the effectiveness of the proposed components.

**Strengths:**

This paper combines diffusion null-space model to the task of LLIE, just like what has been done in the IR tasks, which proves to be valid. Modifications are introduced to better fit the degradation under the Low-light senarios.

**Weaknesses:**

see questions below

**Questions:**

For metric comparison, the improvements are clear. However, for visual comparisons, the improvements are not that clear. For example, in Fig 5,  3rd example, they(Zero-IG, QuadPrior, LightenDiffusion) are quite similar. Authors are suggested to use zoom-in local windows to highlight the superiorities.

As the authors provide a dataset contribution, some more details should be provided regarding the UAV, what is the key advantages of the dataset over previous ones; what type of the scenes are they; under what illumination conditions are they captured.  Clearly demonstrating the advantages is an important way to ensure that the dataset will be used and followed in the future.

The contribution of the "language component" in the framework is not clear.

Evaluations on some unpaired dataset, with perceptual metric LPIPS, could make the paper more persuasive.

---

### Official Review · Reviewer_3fyw · 2024-11-02

**Soundness:** 1
**Presentation:** 3
**Contribution:** 2
**Rating:** 3
**Confidence:** 4

**Summary:**

The manuscript presents an unsupervised approach for low-light image enhancement leveraging a latent consistency model, specifically termed the Range-null Latent Prior-guided Consistency Model (RLPCM). To enhance both consistency and realness, the authors propose to inject language guidance into the noise. Additionally, the authors contribute a new low-light dataset captured using UAVs. Experimental results validate the proposed model’s efficacy.

**Strengths:**

1. The manuscript is logically structured, facilitating ease of comprehension.
2. Ablation studies demonstrated the effectiveness of the proposed design.

**Weaknesses:**

1. In line 320, the manuscript indicates that the lightening decoder is trained using a benchmark dataset. This introduces an unfair advantage over prior methods that adhere to stricter training protocols.
2. The assertion that the null space of matrix A contains only {x = 0} holds true given that A is consistently a full-rank operator. Under this condition, the proposed “null-space refinement” degrade to a classifier-free guidance mechanism, thereby undermining the novelty and technical robustness of the approach.

3. The motivation for introducing a new UAV-captured low-light dataset is ambiguous and insufficiently justified.

4. The novelty of the proposed method is limited. The application of range-null space decomposition within a diffusion-based framework and the use of text prompts for classifier-free guidance are established concepts in low-level vision.

**Questions:**

1. The manuscript suggests training on a benchmark dataset, which raises concerns. Would it be feasible to train on other commonly used, normal-light datasets, such as COCO or ImageNet, similar to prior methods?
2. What is the specific advantage of the newly introduced UAV-LL dataset? The rationale for adding another low-light dataset needs to be clarified, given the existing abundance of low-light image datasets used for training and evaluation..
3. The claim that the null space of A contains only {x = 0} implies that A remains a full-rank operator throughout. How does the proposed “null-space refinement” function effectively when there is no refinement space available in the null space?

---

### Official Review · Reviewer_twM3 · 2024-11-03

**Soundness:** 3
**Presentation:** 3
**Contribution:** 3
**Rating:** 5
**Confidence:** 4

**Summary:**

The paper presents the Range-null Latent Prior-guided Consistency Model (RLPCM), which introduces a novel approach to low-light image enhancement (LLIE) by integrating a latent consistency model (LCM) with Retinex-based range-null space decomposition. This combination of concepts is creative and offers a new perspective on the LLIE problem.

**Strengths:**

The integration of an off-the-shelf LCM as a generative prior to enhance latent consistency and realness in low-light images is a unique contribution to the field. The inclusion of Algorithm 1 provides a clear, step-by-step guide to the RLPCM process, which is beneficial for reproducibility and practical application.  The inclusion of a natural language guidance module to learn the null-space component is a creative application of language models in LLIE tasks.

**Weaknesses:**

1. The global illumination factor assumption mentioned in the paper regarding Range-null Space Decomposition is robustness-deficient, especially in real nighttime environments with diverse lighting sources, such as bustling cityscapes with varied lighting. It seems less credible and should consider local lighting properties rather than relying solely on a constant factor.

2. The qualitative visualization results presented by the authors (as shown in Figures 5 to 12) mainly display night scenes with low contrast and lack exploration of more complex night scenes, such as those in the DARKFACE dataset.

3. In Section 2.3, it is unclear how the global illumination factor, which depends on a constant factor, is calculated. It is also unclear what the various subplots in Figure 4 are intended to illustrate, including the definitions of the axes.

4. In the process of Range-null decomposition, noise removal is not considered. Noise caused by high ISO values or insufficient exposure time is an unavoidable issue for low-light enhancement.

5. The setup of training the Lighting Decoder with an additional 7,000 images collected is unfair when compared to other methods. Why is the decoder's training loss based on Equation 13, in a minimax form?

**Questions:**

1. The method introduces a large number of pre-trained models, and a further assessment is needed regarding computational efficiency compared to related methods, including model parameter count, inference speed, FPS, and training duration.

2. There is a lack of detailed explanation regarding the selection of the pre-trained encoder.

3. The paper does not provide detailed information on the guidance scale parameters used in the natural language guidance mechanism, specifically how these weights were set or chosen in the experiments.

---

### Note · Authors · 2025-03-03

I have read and agree with the venue's withdrawal policy on behalf of myself and my co-authors.

---

### Meta-Review · Area_Chair_aHPT · 2024-12-16

**Metareview:**

The paper was reviewed by five experts and multiple issues were pointed out.

The authors did not provide responses (no rebuttal) to the reviewers and the main concerns remain: unclear motivation, unclear/limited novelty, missing ablations, insufficient experiments, unfair comparisons, missing details.

Therefore, the paper can not be accepted for publication and the authors are invited to benefit from the received feedback and to further improve their work.

**Additional Comments On Reviewer Discussion:**

The authors did not provide responses (no rebuttal) to the reviewers and the main concerns remain: unclear motivation, unclear/limited novelty, missing ablations, insufficient experiments, unfair comparisons, missing details.

---

### Decision · Program_Chairs · 2025-01-22

Reject